# Geometric Feature Embedding for
# Effective 3D Few-Shot Class Incremental Learning

Xiangqi Li [*1 2]  Libo Huang [*1]  Zhulin An [1]  Weilun Feng [1 2]  Chuanguang Yang [1]  Boyu Diao [1]  Fei Wang [1]  Yongjun Xu [1]

## Abstract

3D few-shot class incremental learning (FSCIL) aims to learn new point cloud categories from limited samples while preventing the forgetting of previously learned categories. This research area significantly enhances the capabilities of self-driving vehicles and computer vision systems. Existing 3D FSCIL approaches primarily utilize multimodal pre-trained models to extract the semantic features, heavily dependent on meticulously designed high-quality prompts and fine-tuning strategies. To reduce this dependence, this paper proposes a novel method for **3D F**SCIL with **E**mbedded **G**eometric features (**3D-FLEG**). Specifically, 3D-FLEG develops a point cloud *geometric feature extraction module* to capture category-related geometric characteristics. To address the modality heterogeneity issues that arise from integrating geometric and text features, 3D-FLEG introduces a *geometric feature embedding module*. By augmenting text prompts with spatial geometric features through these modules, 3D-FLEG can learn robust representations of new categories even with limited samples, while mitigating forgetting of the previously learned categories. Experiments conducted on several publicly available 3D point cloud datasets, including ModelNet, ShapeNet, ScanObjectNN, and CO3D, demonstrate 3D-FLEG's superiority over existing state-of-the-art 3D FSCIL methods. Code is available at https://github.com/lixiangqi707/3D-FLEG.

## 1. Introduction

Capturing the 3D shapes of objects from point clouds has become increasingly critical across a wide range of fields, including robotic automation, healthcare, and autonomous driving (Xiao et al., 2024; Xue et al., 2024; Xu et al., 2023; An et al., 2024b;a). In these applications, there is a growing need for models that can continuously adapt to new object characteristics with limited samples while retaining the ability to recognize previously encountered objects (Zhu et al., 2021; Ahmadi et al., 2024; Xing et al., 2025; Huang et al., 2024b; Wu et al., 2025). This growing demand has spurred the development of few-shot class incremental learning (FSCIL) techniques specifically designed for 3D point clouds (Tan & Xiang, 2024; Cheraghian et al., 2025). However, 3D FSCIL faces unique challenges, including catastrophic forgetting, overfitting to new data, and domain gaps between synthetic training data and real-world scans (Tan & Xiang, 2024; Huang et al., 2024a). These challenges significantly increase the complexity of applying FSCIL to 3D point clouds (Chowdhury et al., 2022; Cheraghian et al., 2025).

To address these challenges, Microshape (Chowdhury et al., 2022) proposed a universal description language to mitigate distribution differences. While this approach reduces knowledge forgetting and somewhat addresses data distribution discrepancies, it struggles with learning new categories. Cross-Domain (Tan & Xiang, 2024) introduced distinct recognition methods for new and old categories by incorporating soft and hard-label replay strategies, further alleviating forgetting and enhancing learning capabilities for new categories. Given the extremely limited training sample available for new categories, researchers have explored leveraging pre-trained foundation models to improve learning while minimizing the forgetting of previously learned information (Zhou et al., 2025). The C3PR method investigated how the knowledge from the CLIP model could be applied to 3D point cloud data by projecting point clouds into depth images from optimal angles and integrating a model reprogramming paradigm, offering a new way to handle 3D objects using CLIP (Cheraghian et al., 2025). However, 2D foundation models like CLIP were not de-

---
[*]Equal contribution [1]Institute of Computing Technology, Chinese Academy of Sciences, No. 158 Beiqing Road, Haidian District, Beijing, 100095, China [2]No. 19(A) Yuquan Road, Shijingshan District, Beijing, China, 100049. Correspondence to: Zhulin An <anzhulin@ict.ac.cn>.

signed for 3D tasks and cannot directly process point clouds. Even when depth images are projected and fed into CLIP's image encoder for classification, CLIP's sensitivity to color and texture information leads to lower classification performance due to the absence of these details in the projected images (Wang et al., 2022). In contrast, 3D pre-trained foundation models, which are trained on large datasets of point cloud-image-text pairs, are more adept at handling point cloud data. Ahmadi *et al.* (Ahmadi et al., 2024) were the first to apply 3D foundation models to FSCIL, introducing an adaptive module that requires no additional training and a dual-cache system, significantly improving the capability of 3D vision-language models in incremental learning tasks. However, similar to the 2D counterparts, the performance of 3D pre-trained models heavily relies on high-quality text prompts and elaborate training strategies, especially when working with limited new samples (Sun et al., 2024a).

As shown in the preliminary 3D FSCIL results in Fig. 1, the choice of prompt and training strategies significantly influences the effectiveness of the foundation model during the incremental learning phase. Surprisingly, a simpler prompt template outperforms more complex ones, resulting in an improvement of approximately 6% in recognition accuracy during the final phase. Manually crafting effective, elaborate prompts can be quite challenging, and intricate training strategies may unnecessarily increase model complexity. Based on this observation, we identify two critical challenges that need to be addressed to advance FSCIL on 3D point clouds: minimizing reliance on text prompts and enhancing the model's ability to learn robust feature representations. One promising approach is to embed geometric information into the prompts, an aspect that has not been fully explored in previous studies.

We propose a novel method called **3D F**ew-shot class incremental **L**earning method with **E**mbedded **G**eometric features (**3D-FLEG**). This method aims to reduce dependence on text prompts while improving the model's capacity to learn robust feature representations by incorporating spatial geometric knowledge from 3D point cloud features.

Specifically, 3D-FLEG integrates geometric feature extraction and geometric embedding modules. The geometric feature extraction module is designed to isolate category-specific features and capture object spatial structures. Its core is the construction of dynamic geometric feature projection clusters (DGPC) through clustering and dimensionality reduction. To select cluster centers in real-world point cloud datasets, which often contain noise and complex topologies, we employ spectral clustering (Ng et al., 2001). This technique effectively identifies complex distributions and non-convex clusters in noisy data, ensuring the selection of representative centers for point clouds (Resani et al., 2025). Furthermore, 3D-FLEG utilizes Laplacian eigenmaps to

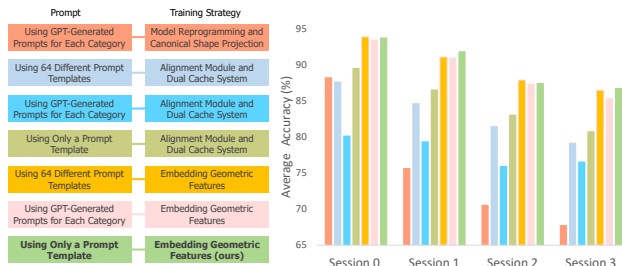

*Figure 1.* Comparison of 3D FSCIL results on "ModelNet → ScanObjectNN" datasets using various 'Prompt-Training Strategy' Combinations: (1) GPT-generated prompts combined with model reprogramming and canonical shape projection (Cheraghian et al., 2025), (2) 64 different prompt templates along with the alignment module and dual cache system (Ahmadi et al., 2024), (3) GPT-generated prompts together with the alignment module and dual cache System, (4) only a prompt template Along with the alignment module and dual cache system, (5) GPT-generated prompts together with embedded geometric features, (6) 64 different prompt templates along with embedded geometric features, and (7) only a prompt template with embedded geometric features.

preserve the local geometric structure during high-to-low dimensional embedding (Belkin & Niyogi, 2003). By capturing both global and local geometric characteristics, this approach boosts the model's representation capabilities while minimizing reliance on text prompts. To further address modality heterogeneity, 3D-FLEG incorporates the geometric feature embedding module. By leveraging multi-head attention mechanisms, this module enables the model to better integrate and utilize the extracted geometric features, thereby enhancing both the model's learning ability and the robustness of its feature representations. We validated 3D-FLEG on the ModelNet, ShapeNet, ScanObjectNN, and CO3D point cloud datasets, demonstrating superior performance in mean accuracy, harmonic mean accuracy, and accuracy drop percentage compared to existing 3D FSCIL methods (Wu et al., 2015; Chang et al., 2015; Uy et al., 2019; Reizenstein et al., 2021; Chowdhury et al., 2022; Tan & Xiang, 2024; Cheraghian et al., 2025; Ahmadi et al., 2024). In summary, our contributions are threefold:

1. We propose 3D-FLEG, a prompt semantic enhancement strategy based on point cloud geometric features. This approach not only mitigates the forgetting of old knowledge but also enables efficient learning of new category representations with minimal training data, thereby providing a novel solution for 3D few-shot class incremental learning.

2. We designed a framework with two innovative modules: a geometric information extraction module that extracts geometric information from point cloud fea-

tures, and a geometric information embedding module that efficiently fuses this information with text features.

3. Our method achieves significant performance improvements in within-dataset and cross-dataset experiments, proving its effectiveness and robustness in handling FSCIL on several publicly used 3D point cloud datasets.

## 2. Related Work

### 2.1. Point Cloud Processing

As one of the primary representations of three-dimensional objects, point clouds have garnered increasing attention due to the growing demand for 3D object representation learning (Xiao et al., 2024). Current approaches to processing point clouds can be primarily categorized into voxel-based methods, projection-based methods, and point-based methods (Lahoud et al., 2022).

Specifically, voxel-based methods map point cloud data into a 3D grid, simplifying computations and supporting convolutional operations. However, they lead to increased computational costs at high resolutions due to higher spatial complexity (Choy et al., 2019; Maturana & Scherer, 2015). Projection-based methods, alternatively, project point clouds onto 2D planes or tangent planes to form multi-view representations, allowing the use of 2D CNNs for feature extraction. While this leverages powerful pattern recognition, it can result in loss of depth information and relies on effective multi-view integration (Huang et al., 2023; Su et al., 2015). In contrast, point-based methods have garnered more attention due to their ability to learn features directly from unordered, variable-length data without transforming them into a regular structure. PointNet pioneered this field by proposing the use of Multi-Layer Perceptrons to independently process each point, combined with global max pooling to generate permutation-invariant features (Qi et al., 2017a). Its successor, PointNet++, further developed hierarchical local feature aggregation strategies, enhancing the capture of local geometric structures (Qi et al., 2017b). To improve feature expressiveness, researchers have drawn inspiration from successful models in the 2D domain, such as CNNs, Graph Neural Networks, and BERT, developing advanced point cloud encoders that significantly boost the capability to understand and process point clouds (Li et al., 2018; Wang et al., 2019; Yu et al., 2022; Liu et al., 2019; Wu et al., 2019). Leveraging these mature point cloud encoders, there has recently been an emergence of foundational 3D models aimed at constructing a unified three-dimensional representation (Zhou et al., 2023; Xue et al., 2023; Liu et al., 2024b; Xue et al., 2024; Zhang et al., 2024; Qi et al., 2024; Zhang et al., 2023). Among these, the Uni3D model utilizes pre-trained Vision Transformers and is trained in an end-to-end manner, achieving effective alignment between 3D point cloud features and image-text features (Zhou et al., 2023). This approach has demonstrated significant potential across a variety of point cloud processing tasks.

### 2.2. Few-Shot Class Incremental Learning

Few-shot class incremental learning has garnered significant attention due to its high challenge and practical research significance (Tao et al., 2020; Cheraghian et al., 2021; Zhu et al., 2021; Peng et al., 2022; Ahmad et al., 2022; Liu et al., 2022; Kang et al., 2022; Kim et al., 2023; Liu et al., 2023; Lin et al., 2024; Sun et al., 2024b; Liu et al., 2024a; Wang et al., 2024; Park et al., 2024; Zheng et al., 2025). Microshapes was the first study to explore FSCIL on 3D point clouds (Chowdhury et al., 2022). It highlighted that FSCIL tasks often rely on extensive synthetic data for training but require continuous learning of new real-world categories, introducing a cross-domain discrepancy challenge. To address this, Microshapes introduced a universal description language to reduce data distribution differences. Meanwhile, Cross-Domain employed a dual-branch architecture and a label replay strategy to mitigate catastrophic forgetting (Tan & Xiang, 2024). Pre-trained foundation models, rich in knowledge, are well-suited to meet FSCIL challenges. The C3PR method explored the application of CLIP models to 3D point cloud processing by generating depth images from optimal projection angles and combining them with model reprogramming paradigms, offering a novel approach (Cheraghian et al., 2025). However, 2D models like CLIP have limitations when directly applied to point clouds, such as sensitivity to color and texture information, which can degrade classification performance. With the emergence of 3D pre-trained models, researchers have shifted towards using models specifically designed for point clouds. FoundationModel was the first to apply a 3D foundation model to FSCIL, introducing an adaptive module that requires no additional training and a dual-cache system, significantly enhancing the ability of 3D vision-language models to handle continual learning tasks (Ahmadi et al., 2024).

Despite their potential to effectively address FSCIL challenges, the performance of 3D pre-trained models still depends on high-quality text prompts and sophisticated training strategies. Therefore, there is a need for new training approaches. These new methods should minimize this dependence and fully unleash the latent capabilities of these models. To this end, we propose a method based on point cloud geometric feature embedding. This approach effectively addresses catastrophic forgetting and cross-domain discrepancies, thereby improving the model's ability to learn effective feature representations from limited new category data.

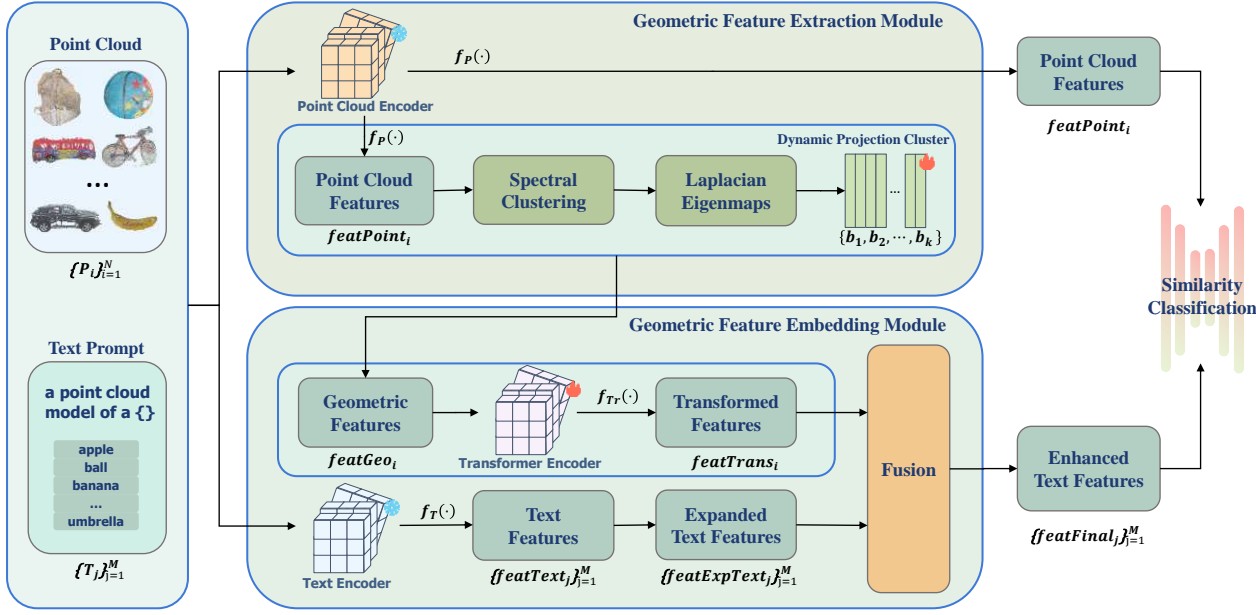

Figure 2. Overall pipeline of 3D-FLEG. The process begins with point cloud data $\{P_i\}_{i=1}^{N}$ and category descriptions formatted using the template "a point cloud model of a { }", denoted as $\{T_j\}_{j=1}^{M}$. A pre-trained frozen 3D encoder extracts initial features from the point cloud data, while a frozen text encoder generates corresponding text features. Next, the model employs spectral clustering and Laplacian eigenmaps to construct dynamic geometric feature projection clusters that are specifically designed to continuously adapt to novel data distributions. Leveraging learnable attention weights for basis vectors, these dynamic projection clusters serve the purpose of extracting geometric information. Subsequently, the extracted geometric features are then processed by a trainable Transformer encoder to generate semantically rich representations. Finally, the model fuses geometric and expanded text features for comprehensive embeddings. During inference, it optimizes parameters by minimizing cross-entropy loss between the original point cloud features and enhanced text features.

## 3. Method

### 3.1. Problem Formulation

3D FSCIL faces a sequence of $T$ tasks, $\mathcal{Q} = \{\mathcal{Q}_1, \mathcal{Q}_2, \ldots, \mathcal{Q}_T\}$, where $\mathcal{Q}_1$ is the base task and $\mathcal{Q}_2, \ldots, \mathcal{Q}_T$ are novel tasks introduced incrementally. Each task $\mathcal{Q}_t$ has a distinct set of classes $\mathcal{C}_t$, ensuring $\mathcal{C}_i \cap \mathcal{C}_j = \emptyset$ for any $i \neq j$. Each class within a task includes a prompt description $\mathcal{P}_t$. Thus, each task $\mathcal{Q}_t$ can be represented as a set of tuples $\{(X_{ti}, y_{ti}, p_{ti})\}_{i=1}^{n_t}$, where:

- $X_{ti} = \{x_{tij}\}_{j=1}^{l_i}$ denotes the $i$-th 3D point cloud with $l_i$ points in $\mathbb{R}^3$.

- $y_{ti} \in \mathcal{C}_t$ is the class label.

- $p_{ti} \in \mathcal{P}_t$ is the class prompt description.

- $n_t$ represents the number of instances per task $\mathcal{Q}_t$.

In 3D FSCIL, the model initially trains on a large synthetic

dataset for $\mathcal{Q}_1$. For subsequent tasks ($t > 1$), training uses limited real-world 3D point clouds per new class. Training proceeds sequentially from $t = 1$ to $t = T$, with the model receiving $X_t$, $y_t$, and prompt descriptions up to the current task $\{\mathcal{P}_1, \ldots, \mathcal{P}_t\}$. After training on $\mathcal{Q}_t$, the model must classify test samples from all tasks up to $\mathcal{Q}_t$, ensuring it retains knowledge of previous classes while learning new ones. This formulation evaluates the model's ability to handle incremental learning challenges, especially when new classes have limited data, balancing old knowledge retention and new information acquisition (Chowdhury et al., 2022; Cheraghian et al., 2025).

### 3.2. Model Overview

Our proposed method, 3D-FLEG, is illustrated in Fig. 2. It begins with data preparation and initialization and takes as input point cloud data $\{P_i\}_{i=1}^{N}$ and a list of categories with predefined templates $\{T_j\}_{j=1}^{M}$, where $N$ is the number of point clouds, $M$ is the number of categories. A pre-

trained 3D base model's point cloud encoder $f_P(\cdot)$ extracts initial features $\{featBase_i\}_{i=1}^N = \{f_P(P_i)\}_{i=1}^N$, while a text encoder $f_T(\cdot)$ generates corresponding text features $\{featText_j\}_{j=1}^M$. Subsequently, the model constructs dynamic geometric feature projection clusters for the base-class point clouds. Initial features are clustered using spectral clustering combined with K-means, followed by dimensionality reduction via Laplacian eigenmap to compute basis vectors $B = \{b_1, b_2, ..., b_k\}$. Upon the arrival of new classes, these clusters adapt to new distributions, with each basis vector $b_i$ associated with a learnable weight $w_i$. Point cloud features $featPoint_i$ are then dot-multiplied with basis vectors and aggregated through a weighted sum to extract geometric information $featGeo_i$. In the geometric feature embedding module, geometric features are processed by a Transformer encoder with self-attention layers to generate semantically enriched feature representations $featTrans_i$. Text features are integrated by concatenating $featTrans_i$ with expanded, dimension-matched text features $\{featExpText_j\}_{j=1}^M$, resulting in the final fused feature $\{featFinal_j\}_{j=1}^M$.

During inference, 3D-FLEG compares original point cloud features $featPoint_i$ with the enhanced text features $\{featFinal_j\}_{j=1}^M$, guided by cross-entropy loss $\mathcal{L}$ for optimization. By minimizing this loss, the model refines its parameters to better accommodate evolving data distributions and improve classification accuracy.

### 3.3. Geometric Feature Extraction Module

In 3D-FLEG, we introduce a geometric feature extraction module designed to extract more representative and stable high-level geometric attributes from raw point cloud features, serving the task of text enhancement. This module first uses spectral clustering to select cluster centers from the base-class point cloud features and then applies generalized eigenmaps for dimensionality reduction, generating dynamic projection clusters.

For base-class point cloud data, we extract an initial set of point cloud features $\{featBase_i\}_{i=1}^N = \{f_P(P_i)\}_{i=1}^N$. Spectral clustering on $featBase$ yields cluster assignments $C$, with K-means applied internally for final clustering. The affinity matrix $W$ is constructed as follows:

$$W_{ij} = \exp\left(-\frac{\|x_i - x_j\|^2}{2\sigma^2}\right).$$ (1)

The degree matrix $D$ has diagonal elements given by:

$$D_{ii} = \sum_j W_{ij}.$$ (2)

Normalization of the Laplacian matrix $L$ is performed as:

$$L = I - D^{-\frac{1}{2}} W D^{-\frac{1}{2}}.$$ (3)

Eigendecomposition on $L$ selects the top $k$ smallest non-zero eigenvalues' corresponding eigenvectors to form matrix $U$. Each row of $U$ is normalized to obtain the final feature representation $Y = [y_1, y_2, ..., y_k]^T$. K-means clustering on $Y$ achieves the final clustering result $C$.

For Laplacian eigenmap dimensionality reduction, a set of basis vectors $B = \{b_1, b_2, ..., b_k\}$ is calculated from the clustering results. The generalized eigenvalue problem $Lz = \lambda Dz$ is solved, where $z$ represents eigenvectors and $\lambda$ represents eigenvalues. Eigenvectors corresponding to the top $k$ smallest eigenvalues form a matrix $Z$, which is standardized to obtain the basis vector matrix $B$.

When new classes arrive, the geometric feature projection cluster $B$ is updated based on an update rate parameter to adapt to new distributions. For test samples, each basis vector $b_i \in B$ has a learnable weight $w_i \in W$. Geometric information is aggregated via a weighted sum to get geometric features featGeo$_i$:

$$\text{featGeo}_i = \sum_{j=1}^k w_j \cdot (\text{featPoint}_i \cdot b_j).$$ (4)

Spectral clustering, based on spectral analysis from graph theory, is well-suited for capturing data distributions with complex topological structures and is robust to noise. Laplacian eigenmaps, as a manifold learning technique, preserve the local geometric properties of the point clouds. By computing the weighted similarity between point cloud features and dynamic projection clusters, we extract geometric features that exhibit good invariance to different viewpoints and scale transformations.

### 3.4. Geometric Feature Embedding Module

For 3D FSCIL, we have designed a novel geometric feature embedding module aimed at fully utilizing the rich spatial and geometric information from point cloud data and the semantic information from text descriptions. This module optimizes the processing of geometric features to ensure that data from both modalities interact on similar levels of abstraction, thereby achieving more effective information integration. Specifically, 3D-FLEG incorporates both AdaptiveAvgPool1d and AdaptiveMaxPool1d to extract features at different granularity levels. In our design, we leverage these pooling methods to effectively capture both fine-grained and coarse-grained characteristics of the input data:

$$F_p = [AvgPool(featGeo_i), MaxPool(featGeo_i)].$$ (5)

Here, $F_p$ combines both levels of detail from the geometric features. Next, $F_p$ undergoes linear projection and is passed through a Transformer-based multi-head attention mechanism to capture long-range dependencies and optimize

feature relevance. The resulting features are further processed through a fully connected layer to adjust the feature space. These transformations refine the geometric features and prepare them for fusion with text features. We denote the final transformed feature as $featTrans_i$.

$$featTrans_i = f_{Tr}(F_p), \qquad (6)$$

where $f_{Tr}$ denotes the processing through a Transformer encoder. The fused feature $featFinal_j$ is obtained by concatenating processed geometric features $featTrans_i$ with dimension-matched expanded text features $featExpText_j$, followed by an activation function:

$$featFinal_j = \tanh\left([featTrans_i, featExpText_j]\right), \quad (7)$$

where $featExpText_j$ refers to text features expanded to align with the geometric feature dimensions, facilitating their effective fusion.

This structured approach ensures effective transformation and integration of geometric features with textual information, leading to improved model performance and adaptability.

### 3.5. Training Pipeline

During model training, both the point cloud encoder and the text encoder remain frozen. Only the dynamic geometric feature projection clusters in the geometric feature extraction module, the weights of each basis vector within them, and the parameters of the geometric feature embedding module are trained. 3D-FLEG primarily relies on two key components: the geometric feature extraction module and the geometric feature embedding module. In the geometric feature extraction module, the core step involves constructing initial dynamic geometric feature projection clusters using base-class data. Notably, during the incremental learning phase, we introduce an update rate to adjust the dynamic projection clusters and train the corresponding weights for each basis vector to adapt to the new data distribution. For each input sample, the dynamic geometric feature projection clusters extract category-relevant geometric features from the raw point cloud features. Subsequently, these geometric features are transmitted to the geometric feature embedding module for semantic enhancement of text features. Finally, 3D-FLEG is trained using a loss function that computes the categorical cross-entropy loss between the original point cloud features and the enhanced text features:

$$\mathcal{L} = -\frac{1}{N}\sum_{i=1}^{N}\sum_{k=1}^{K} g_{ik} \log(\hat{q}_{ik}). \qquad (8)$$

where $\mathbf{g}_{ik}$ denotes the one-hot encoded ground-truth label vector, and $\hat{q}_{ik}$ represents the predicted class probability distribution generated from the fused features $\{featFinal_j\}_{j=1}^{M}$.

## 4. Experiment

### 4.1. Dataset Partitioning and Evaluation Metrics

**Dataset partitioning.** For dataset partitioning, we first conducted within-dataset experiments to establish baseline performance. Using ModelNet, we allocated 20 base classes with the remaining 20 classes divided into four incremental stages. For ShapeNet and CO3D, we utilized 25 base classes, distributing the incremental classes across 7 or 6 tasks respectively, comprising either 30 or 25 incremental classes. To simulate limited real-scanned data for new categories, we also designed cross-dataset experiments. For the transition from ModelNet to ScanObjectNN, we followed (Chowdhury et al., 2022) with four tasks. In the case of ShapeNet to ScanObjectNN, we structured four tasks involving 44 ShapeNet base classes and 15 ScanObjectNN incremental classes. Lastly, for ShapeNet to CO3D, we set up eleven tasks with 44 ShapeNet base classes and 50 CO3D incremental classes, representing the most challenging setup.

**Evaluation metrics.** In our experiment, we use three key metrics:

- **Average Accuracy**: We calculate overall accuracy after each incremental step, covering both base and new classes. This measures the proportion of correctly classified samples out of the total number of samples.

$$\text{Average Accuracy} = \frac{\text{Correct samples}}{\text{Total samples}}. \qquad (9)$$

- **Relative Accuracy Drop Rate** ($\Delta$): We introduce $\Delta$ to quantify performance changes during incremental learning:

$$\Delta = \left|\frac{\text{acc}_T - \text{acc}_0}{\text{acc}_0}\right| \times 100, \qquad (10)$$

where $\text{acc}_T$ and $\text{acc}_0$ are the accuracies of the last and first incremental tasks, respectively. Lower $\Delta$ values indicate better stability.

- **Harmonic Accuracy** ($A_h$): To balance performance on old and new classes, especially given the limited data for new classes, we adopt harmonic accuracy:

$$A_h = \frac{2 \times A_b \times A_n}{A_b + A_n}, \qquad (11)$$

where $A_b$ and $A_n$ are the accuracies of base and new classes, respectively. Higher $A_h$ indicates a better balance between old and new class performances. Additionally, we report separate accuracies for base and new classes at each incremental stage for detailed analysis (Peng et al., 2022).

Table 1. Average accuracy within a single dataset.

| Method | ModelNet | | | | | | CO3D | | | | | | | ShapeNet | | | | | | | |
|---|---|---|---|---|---|---|---|---|---|---|---|---|---|---|---|---|---|---|---|---|---|
| | 20 | 25 | 30 | 35 | 40 | Δ↓ | 25 | 30 | 35 | 40 | 45 | 50 | Δ↓ | 25 | 30 | 35 | 40 | 45 | 50 | 55 | Δ↓ |
| FT | 89.8 | 9.7 | 4.3 | 3.3 | 3.0 | 96.7 | 76.7 | 11.2 | 3.6 | 3.2 | 1.8 | 0.8 | 99.0 | 87.0 | 25.7 | 6.8 | 1.3 | 0.9 | 0.6 | 0.4 | 99.5 |
| Joint | 89.8 | 88.2 | 87.0 | 83.5 | 80.5 | 10.4 | 76.7 | 69.4 | 64.8 | 62.7 | 60.7 | 59.8 | 22.0 | 87.0 | 85.2 | 84.3 | 83.0 | 82.5 | 82.2 | 81.3 | 6.6 |
| LwF | 89.8 | 36.0 | 9.1 | 3.6 | 3.1 | 96.0 | 76.7 | 14.7 | 4.7 | 3.5 | 2.3 | 1.0 | 98.7 | 87.0 | 60.8 | 33.5 | 15.9 | 3.8 | 3.1 | 1.8 | 97.9 |
| IL2M | 89.8 | 65.5 | 58.4 | 52.3 | 53.6 | 40.3 | 76.7 | 31.5 | 27.7 | 18.1 | 27.1 | 21.9 | 71.4 | 87.0 | 58.6 | 45.7 | 40.7 | 50.1 | 49.4 | 49.3 | 43.3 |
| ScaIL | 89.8 | 66.8 | 64.5 | 58.7 | 56.5 | 37.1 | 76.7 | 39.5 | 34.1 | 24.1 | 30.1 | 27.5 | 64.1 | 87.0 | 56.6 | 51.8 | 44.3 | 50.3 | 46.3 | 45.4 | 47.8 |
| EEIL | 89.8 | 75.4 | 67.2 | 60.1 | 55.6 | 38.1 | 76.7 | 61.4 | 52.4 | 42.8 | 39.5 | 32.8 | 57.2 | 87.0 | 77.7 | 73.2 | 69.3 | 66.4 | 65.9 | 65.8 | 22.4 |
| FACT | 90.4 | 81.3 | 77.1 | 73.5 | 65.0 | 28.1 | 77.9 | 67.1 | 59.7 | 54.8 | 50.2 | 46.7 | 40.0 | 87.5 | 75.3 | 71.4 | 69.9 | 67.5 | 65.7 | 62.5 | 28.6 |
| Sem-aware | 91.3 | 82.2 | 74.3 | 70.0 | 64.7 | 29.1 | 78.6 | 66.9 | 59.2 | 53.6 | 49.1 | 42.9 | 44.1 | 87.2 | 74.9 | 68.1 | 69.0 | 68.1 | 66.9 | 63.8 | 26.8 |
| Microshape | 93.6 | 83.1 | 78.2 | 75.8 | 67.1 | 28.3 | 78.5 | 67.3 | 60.1 | 56.1 | 51.4 | 47.2 | 39.9 | 87.6 | 83.2 | 81.5 | 79.0 | 76.8 | 73.5 | 72.6 | 17.1 |
| C3PR | 91.6 | 82.3 | 75.8 | 72.2 | 70.9 | 22.5 | 81.5 | 69.4 | 66.5 | 63.0 | 54.2 | 53.8 | 34.0 | 88.0 | 81.6 | 77.8 | 76.7 | 76.9 | 76.2 | 74.7 | 15.1 |
| **3D-FLEG(ours)** | **98.3** | **95.5** | **93.5** | **91.4** | **87.3** | **11.2** | **82.1** | **75.4** | **69.7** | **66.7** | **57.7** | **57.3** | **30.2** | **94.3** | **91.4** | **89.7** | **89.2** | **88.4** | **86.2** | **83.0** | **12.0** |

Table 2. Average accuracy across datasets.

| Method | ShapeNet → CO3D | | | | | | | | | | | | ModelNet → ScanObjectNN | | | | | ShapeNet → ScanObjectNN | | | | |
|---|---|---|---|---|---|---|---|---|---|---|---|---|---|---|---|---|---|---|---|---|---|---|
| | 39 | 44 | 49 | 54 | 59 | 64 | 69 | 74 | 79 | 84 | 89 | Δ↓ | 26 | 30 | 34 | 37 | Δ↓ | 44 | 49 | 54 | 59 | Δ↓ |
| FT | 81.0 | 20.2 | 2.3 | 1.7 | 0.8 | 1.0 | 1.0 | 1.3 | 0.9 | 0.5 | 1.6 | 98.0 | 88.4 | 6.4 | 6.0 | 1.9 | 97.9 | 81.4 | 38.7 | 4.0 | 0.9 | 98.9 |
| Joint | 81.0 | 79.5 | 78.3 | 75.2 | 75.1 | 74.8 | 72.3 | 71.3 | 70.0 | 68.8 | 67.3 | 16.9 | 88.4 | 79.7 | 74.0 | 71.2 | 19.5 | 81.4 | 82.5 | 79.8 | 78.7 | 3.3 |
| LwF | 81.0 | 57.4 | 19.3 | 2.3 | 1.0 | 0.9 | 0.8 | 1.3 | 1.1 | 0.8 | 1.9 | 97.7 | 88.4 | 35.8 | 5.8 | 2.5 | 97.2 | 81.4 | 47.9 | 14.0 | 5.9 | 92.8 |
| IL2M | 81.0 | 45.6 | 36.8 | 35.1 | 31.8 | 33.3 | 34.0 | 31.5 | 30.6 | 32.3 | 30.0 | 63.0 | 88.4 | 58.2 | 52.9 | 52.0 | 41.2 | 81.4 | 53.2 | 43.9 | 45.8 | 43.7 |
| ScaIL | 81.0 | 50.1 | 45.7 | 39.1 | 39.0 | 37.9 | 38.0 | 36.0 | 33.7 | 33.0 | 35.2 | 56.5 | 88.4 | 56.5 | 55.9 | 52.9 | 40.2 | 81.4 | 49.0 | 46.7 | 40.0 | 50.9 |
| EEIL | 81.0 | 75.2 | 69.3 | 63.2 | 60.5 | 57.9 | 53.0 | 51.9 | 51.3 | 47.8 | 47.6 | 41.2 | 88.4 | 70.2 | 61.0 | 56.8 | 35.7 | 81.4 | 74.5 | 69.8 | 63.4 | 22.1 |
| FACT | 81.4 | 76.0 | 70.3 | 68.1 | 65.8 | 63.5 | 63.0 | 60.1 | 58.2 | 57.5 | 55.9 | 31.3 | 89.1 | 72.5 | 68.3 | 63.5 | 28.7 | 82.3 | 74.6 | 69.9 | 66.8 | 18.8 |
| Sem-aware | 80.6 | 69.5 | 66.5 | 62.9 | 63.2 | 63.0 | 61.2 | 58.3 | 58.1 | 57.2 | 55.2 | 31.6 | 88.5 | 73.9 | 67.7 | 64.2 | 27.5 | 81.3 | 70.6 | 65.2 | 62.9 | 22.6 |
| Microshape | 82.6 | 77.9 | 73.9 | 72.7 | 67.7 | 66.2 | 65.4 | 63.4 | 60.6 | 58.1 | 57.1 | 30.9 | 89.3 | 73.2 | 68.4 | 65.1 | 27.1 | 82.5 | 74.8 | 71.2 | 67.1 | 18.7 |
| C3PR | 83.6 | 80.0 | 77.8 | 75.4 | 72.8 | 72.3 | 70.3 | 67.9 | 64.9 | 64.1 | 63.2 | 24.4 | 88.3 | 75.7 | 70.6 | 67.8 | 23.2 | 84.5 | 77.8 | 75.5 | 71.9 | 14.9 |
| FoundationModel | 87.3 | 86.2 | 84.4 | 82.2 | 80.7 | 79.6 | 78.2 | 76.8 | 76.1 | 74.5 | 72.6 | 16.8 | 87.7 | 84.7 | 81.5 | 79.2 | 9.7 | 90.8 | 86.5 | 86.4 | 85.6 | 5.7 |
| **3D-FLEG(ours)** | **91.7** | **90.5** | **89.0** | **86.8** | **84.9** | **83.3** | **82.3** | **81.1** | **79.8** | **77.8** | **76.8** | **16.2** | **93.8** | **91.9** | **87.5** | **86.8** | **7.5** | **90.9** | **89.1** | **87.1** | **86.3** | **5.1** |

## 4.2. Implementation Details

In the implementation of the incremental stages, we randomly selected five samples per category and retained one sample from previously learned categories, simulating practical data scarcity. The dynamic geometric feature projection cluster contains 1024 base vectors and has an update rate of 0.1 during the incremental phase. For all samples, we select 1024 points from the 3D point cloud objects using the farthest point sampling method as the input. Specifically, considering the computational overhead and model performance, our experiment is configured the same as the (Ahmadi et al., 2024). We employed the "EVA02-E-14+" CLIP model and the "eva02-base_patch14_448" model as our point cloud encoder. The Transformer encoder comprises 2 standard layers, each with 8-head self-attention. As indicated by Uni3D, while more training parameters can lead to better training results, it also incurs greater computational overhead (Zhou et al., 2023). Therefore, we opted for the basic version of the point cloud encoder. Additionally, the entire experimental process was conducted on a single NVIDIA A100 GPU. The optimizer utilized is the AdamW optimizer, with the weight decay set to $1 \times 10^{-4}$. During the basic category training, we trained for 10 epochs with a learning rate of 0.0005. For the new categories, we increased the training to 50 epochs and set the learning rate to 0.001, maintaining a fixed batch size of 32.

## 4.3. Overall Experiment

**Compared methods.** In this section, we evaluate a series of methods for FSCIL on 3D point clouds. These methods encompass fine-tuning and joint training, where fine-tuning updates the model using only data from new categories to simplify the incremental learning process, while joint training allows the model to simultaneously access data from all categories to mitigate forgetting issues. Additionally, we adapt common 2D FSCIL methods such as IL2M, ScaIL, EEIL, LwF, and FACT by replacing their CNNs with PointNet to accommodate point cloud data, exploring the applicability of established 2D techniques in 3D contexts(Belouadah & Popescu, 2019; 2020; Castro et al., 2018; Li & Hoiem, 2017; Zhou et al., 2022; Cheraghian et al., 2021). Finally, we examine specialized strategies specifically designed for 3D FSCIL tasks, addressing unique challenges associated with point cloud data (Chowdhury et al., 2022; Tan & Xiang, 2024; Cheraghian et al., 2025; Ahmadi et al., 2024).

**Experimental results and analysis.** In Tab. 1 and Tab. 2, we present the experimental comparison results for within-

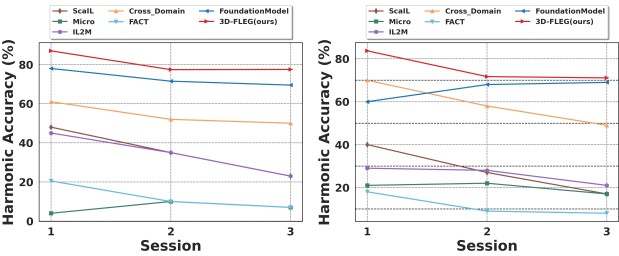

*Figure 3.* Experimental results of harmonic accuracy on ModelNet → ScanObjectNN (Left) and ShapeNet → ScanObjectNN (Right).

dataset and cross-dataset scenarios. [1] These results indicate that while many existing incremental learning methods perform admirably with 2D images, they tend to exhibit suboptimal performance when applied to 3D point cloud data. This shortfall primarily stems from their inability to effectively address catastrophic forgetting and the distributional shifts between different data sources. In contrast, methods specifically designed for few-shot incremental learning on point clouds, such as Microshape, demonstrate superior performance. By leveraging the powerful pre-training knowledge of the base model, C3PR and FoundationModel have also shown enhanced capabilities in mitigating forgetting and learning new categories. Our proposed method, 3D-FLEG, advances this further by seamlessly integrating geometric features of point clouds with text prompts, thereby augmenting the capabilities of 3D foundation models. In experiments conducted within a single dataset, 3D-FLEG not only demonstrates a significantly smaller drop in accuracy but also achieves an approximate 4% increase in accuracy over existing state-of-the-art methods at each incremental stage. In the more challenging cross-dataset experiments, our approach also significantly outperforms the existing optimal solutions and achieves the best performance.

It is worth noting that since the absolute values of the average accuracy recorded in the tables may be affected by the training effect of the base model, it cannot comprehensively measure the specific performance of the model during the incremental stage. To more accurately reflect the forgetting situation of the model during the incremental learning stage, we further calculated the accuracy drop of each incremental stage compared to the base model. For detailed information, please refer to Tab. 5 and Tab. 6 in the Appendix A.

In addition, to further highlight the accuracy of our method in classifying new categories, we adopted the harmonic

---

[1]The experimental results for the comparison methods presented in the table are derived from the studies in (Ahmadi et al., 2024) and (Cheraghian et al., 2025), with all experiments conducted under consistent settings. Note that the FoundationModel was excluded from Tab. 1 because its original publication did not report results within the dataset, and notable performance discrepancies were encountered during our reproduction attempts.

*Table 3.* Effectiveness of the dynamic geometric feature projection clusters and attention weight in 3D-FLEG.

| DGPC | Attention Weight | Average Accuracy (%) | | | | Harmonic Accuracy (%) | | |
|---|---|---|---|---|---|---|---|---|
| | | 0 | 1 | 2 | 3 | 1 | 2 | 3 |
| ✗ | ✗ | **93.8** | 89.5 | 83.8 | 84.1 | 85.1 | 75.4 | 76.2 |
| ✓ | ✗ | **93.8** | 91.6 | 86.7 | 84.9 | 86.4 | 76.5 | 75.4 |
| ✓ | ✓ | **93.8** | **91.9** | **87.5** | **86.8** | **87.0** | **77.4** | **77.5** |

mean metric evaluation index and conducted experiments on two cross-datasets. As shown in Fig. 3, 3D-FLEG outperforms current state-of-the-art methods in all scenarios, which fully demonstrates that it can not only better reduce the forgetting of old knowledge but also efficiently learn the feature representations of new classes using limited samples.

### 4.4. Ablation Study

To evaluate the effectiveness of our proposed method and its individual modules, we conducted a series of ablation studies on the cross-dataset task from ModelNet to ScanObjectNN. During the evaluation, we utilized average accuracy and harmonic accuracy as our metrics.

**Evaluating component effectiveness in 3D-FLEG.** To rigorously evaluate the effectiveness of each component in 3D-FLEG, we conducted a series of detailed experiments from three perspectives: assessing the efficacy of the geometric feature extraction module based on dynamic projection clusters, validating the approach for constructing dynamic projection clusters, and evaluating the performance of the geometric feature embedding module.

As illustrated in Tab. 3, incorporating the dynamic geometric feature projection clusters significantly enhances the model's ability to integrate point cloud and textual features, markedly outperforming direct fusion of raw features, which is shown as the first row in the table. By introducing an attention mechanism that assigns differential attention weights to each basis vector within the projection cluster, 3D-FLEG further refines feature representation. Learnable weights prioritize cluster centers relevant to incremental tasks, ensuring that the most pertinent geometric features are emphasized, improving robustness and adaptability. This approach enhances stability in the presence of noise, outliers, and viewpoint changes. Furthermore, we substantiated the effectiveness of the implementation approach for the geometric feature extraction module, and the results are illustrated in Fig. 4. In 3D-FLEG, we employed spectral clustering and Laplacian eigenmaps to process point cloud features. Experiments demonstrated that the dynamic projection clusters constructed by combining these two methods exhibit superior performance in 3D few-shot class incremental learning. To verify the effectiveness and efficiency of the proposed geometric feature embedding module, we conducted ablation studies by replacing it with long short-term memory

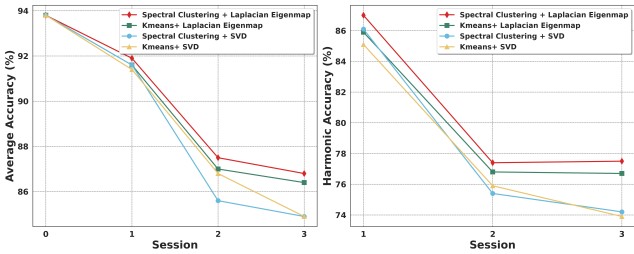

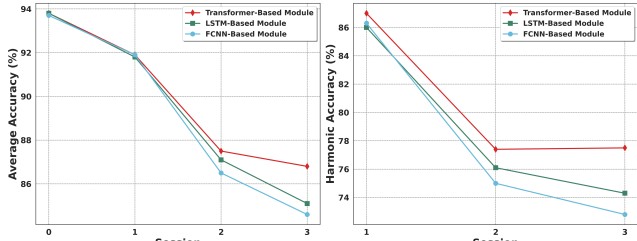

*Figure 4.* Evaluating spectral clustering and Laplacian eigenmaps for geometric feature extraction module.

*Figure 5.* Validation of the geometric feature embedding module.

*Table 4.* Impact of Basis Vector Count in Dynamic Projection Clusters on Experimental Results.

| Basis Vectors Count | Average Accuracy (%) | | | | Harmonic Accuracy (%) | | |
|---|---|---|---|---|---|---|---|
| | 0 | 1 | 2 | 3 | 1 | 2 | 3 |
| 256 | 93.4 | 91.5 | 86.4 | 85.0 | 86.8 | 76.2 | 76.8 |
| 512 | 93.2 | 91.5 | 86.3 | 85.9 | 87.0 | 76.2 | 77.1 |
| 1024 | 93.8 | 91.9 | 87.5 | 86.8 | 87.0 | 77.4 | 77.5 |
| 2048 | 93.6 | 91.6 | 87.5 | 86.7 | 87.3 | 77.7 | 77.7 |
| 4096 | 93.8 | 91.9 | 87.8 | 86.4 | 87.0 | 77.7 | 78.3 |

networks (LSTM) or full connect neural network (FCNN). As shown in Fig. 5, our approach achieved better geometric feature embedding results. By introducing local and global pooling along with Transformer-based modules, our method optimizes feature relevance and reduces redundancy, minimizing unnecessary noise interference. This enables the model to learn more meaningful representations.

**Hyperparameter sensitivity.** In our method, the number of basis vectors contained in the dynamic projection cluster and the update rate of the dynamic projection cluster in the incremental phase are two key parameters. We further explored the impact of different parameters on the experimental results through experiments.

In Tab. 4, we investigate the impact of the number of basis vectors in dynamic projection clusters on the experimental results. The experimental results show that when the number increases to 1024, the performance of the model in the basic stage and the incremental stage tends to be stable. This phenomenon can be attributed to the fact that the base model we use has a feature space of 1024 dimensions. When the number of basis vectors does not align with this feature dimension, additional modules must be introduced to compress or expand the original features to match the required dimensionality. However, this process not only increases model complexity but can also lead to the loss of critical information within the point cloud features or the introduction of redundant information. Therefore, in our experimental setup, to maintain consistency and integrity in feature representation, the number of basis vectors is fixed

at 1024. In addition, we further conducted experiments to investigate the impact of dynamically adjusting the update rate of projection clusters during the incremental learning phase on model performance. Detailed analyses are provided in the Appendix B.

## 5. Conclusion

This paper proposes 3D-FLEG, a novel solution to the challenging and practical problem of few-shot class incremental learning on 3D point clouds. Previous studies leverage well-designed prompts and complex strategies to extract the pre-training knowledge, neglecting the inherent geometric features embedded in the point clouds. To address these issues, 3D-FLEG introduces the geometric feature extraction module and the geometric feature embedding module. These two modules work synergistically to build a text semantic enhancement strategy based on point cloud geometric features, enabling 3D-FLEG to reduce reliance on text prompts and enhance the model's capacity to learn robust feature representations while mitigating the forgetting problem. Experiments on four widely used 3D datasets demonstrate that 3D-FLEG outperforms existing methods, validating its effectiveness and robustness. Despite the excellent performance of 3D-FLEG in 3D FSCIL, further advanced geometric feature extraction and embedding methods are worth exploring. Extending 3D-FLEG to broader 3D vision applications, *i.e.,* 3D object detection and scene segmentation, is also attractive to the community.

## Acknowledgments

This work is partially supported by the Beijing Natural Science Foundation under Grant Number 4244098, the National Natural Science Foundation of China under Grant Number 62476264 and 62406312, the Postdoctoral Fellowship Program and China Postdoctoral Science Foundation under Grant Number BX20240385 (China National Postdoctoral Program for Innovative Talents), and the Science Foundation of the Chinese Academy of Sciences.

## Impact Statement

3D-FLEG provides a powerful new approach for 3D few-shot class incremental learning, not only advancing relevant technologies in the field of machine learning but also significantly boosting applications that rely on 3D data analysis, such as medical imaging, autonomous driving technologies, and virtual reality. In the long term, 3D-FLEG is expected to foster continuous innovation in these fields, enhancing the quality and efficiency of products and services based on 3D data.

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

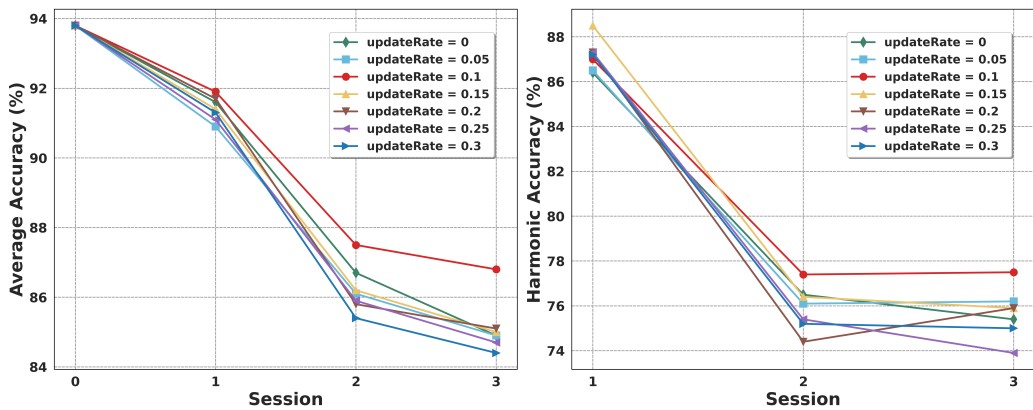

*Figure 6.* Impact of dynamic projection cluster update rate during incremental learning phase on model performance.

*Table 5.* Relative accuracy drop rate within a single dataset

| Method | ModelNet | | | | | | CO3D | | | | | | | ShapeNet | | | | | | | |
|--------|----|----|----|----|----|------|----|----|----|----|----|----|------|----|----|----|----|----|----|----|------|
| | 20 | 25 | 30 | 35 | 40 | Mean | 25 | 30 | 35 | 40 | 45 | 50 | Mean | 25 | 30 | 35 | 40 | 45 | 50 | 55 | Mean |
| FT | - | 89.2 | 95.2 | 96.3 | 96.7 | 94.4 | - | 85.4 | 95.3 | 95.8 | 97.7 | 99.0 | 94.6 | - | 70.5 | 92.2 | 98.5 | 99.0 | 99.3 | 99.5 | 93.2 |
| Joint | - | 1.8 | 3.1 | 7.0 | 10.4 | 5.6 | - | 9.5 | 15.5 | 18.3 | 20.9 | 22.0 | 17.2 | - | 2.1 | 3.1 | 4.6 | 5.2 | 5.5 | 6.6 | 4.5 |
| LwF | - | 59.9 | 89.9 | 96.0 | 96.5 | 85.6 | - | 80.8 | 93.9 | 95.4 | 97.0 | 98.7 | 93.2 | - | 30.1 | 61.5 | 81.7 | 95.6 | 96.4 | 97.9 | 77.2 |
| IL2M | - | 27.1 | 35.0 | 41.8 | 40.3 | 36.0 | - | 58.9 | 63.9 | 76.4 | 64.7 | 71.4 | 67.1 | - | 32.6 | 47.5 | 53.2 | 42.4 | 43.2 | 43.3 | 43.7 |
| ScaIL | - | 25.6 | 28.2 | 34.6 | 37.1 | 31.4 | - | 48.5 | 55.5 | 68.6 | 60.8 | 64.1 | 59.5 | - | 34.9 | 40.5 | 49.1 | 42.2 | 46.8 | 47.8 | 43.6 |
| EEIL | - | 16.0 | 25.2 | 33.1 | 38.1 | 28.1 | - | 19.9 | 31.7 | 44.2 | 48.5 | 57.2 | 40.3 | - | 10.7 | 15.9 | 20.3 | 23.7 | 24.3 | 24.4 | 19.9 |
| FACT | - | 10.1 | 14.7 | 18.7 | 28.1 | 17.9 | - | 13.9 | 23.4 | 29.7 | 35.6 | 40.1 | 28.5 | - | 13.9 | 18.4 | 20.1 | 22.9 | 24.9 | 28.6 | 21.5 |
| Sem-aware | - | 10.0 | 18.6 | 23.3 | 29.1 | 20.2 | - | 14.9 | 24.7 | 31.8 | 37.5 | 45.4 | 30.9 | - | 14.1 | 21.9 | 20.9 | 21.9 | 23.3 | 26.8 | 21.5 |
| Microshape | - | 11.2 | 16.5 | 19.0 | 28.3 | 18.8 | - | 14.3 | 23.4 | 28.5 | 34.5 | 39.9 | 28.1 | - | 5.0 | 7.0 | 9.8 | 12.3 | 16.1 | 17.1 | 11.2 |
| C3PR | - | 10.2 | 17.2 | 21.2 | 22.6 | 17.8 | - | 14.8 | 18.4 | 22.7 | 33.5 | 34.0 | 24.7 | - | 7.3 | 11.6 | 12.8 | 12.6 | 13.4 | 15.1 | 12.1 |
| **3D-FLEG(ours)** | - | **2.8** | **4.9** | **7.0** | **11.2** | **6.5** | - | **8.2** | **15.1** | **18.8** | **29.7** | **30.2** | **20.4** | - | **3.1** | **4.9** | **5.4** | **6.3** | **8.6** | **12.0** | **6.7** |

*Table 6.* Relative accuracy drop rate across datasets.

| Method | ShapeNet → CO3D | | | | | | | | | | | | ModelNet → ScanObjectNN | | | | | ShapeNet → ScanObjectNN | | | | |
|--------|----|----|----|----|----|----|----|----|----|----|----|------|----|----|----|----|------|----|----|----|----|------|
| | 39 | 44 | 49 | 54 | 59 | 64 | 69 | 74 | 79 | 84 | 89 | Mean | 26 | 30 | 34 | 37 | Mean | 44 | 49 | 54 | 59 | Mean |
| FT | - | 75.1 | 97.2 | 97.9 | 99.0 | 98.8 | 98.8 | 98.4 | 98.9 | 99.4 | 98.0 | 96.2 | - | 92.8 | 93.2 | 97.9 | 94.6 | - | 52.5 | 95.1 | 98.9 | 82.2 |
| Joint | - | 1.9 | 3.3 | 7.2 | 7.3 | 7.7 | 10.7 | 12.0 | 13.6 | 15.1 | 16.9 | 9.6 | - | 9.8 | 16.3 | 19.5 | 15.2 | - | -1.4 | 2.0 | 3.3 | 1.3 |
| LwF | - | 29.1 | 76.2 | 97.2 | 98.8 | 98.9 | 99.0 | 98.4 | 98.6 | 99.0 | 97.7 | 89.3 | - | 59.5 | 93.4 | 97.2 | 83.4 | - | 41.2 | 82.8 | 92.8 | 72.3 |
| IL2M | - | 43.7 | 54.6 | 56.7 | 60.7 | 58.9 | 58.0 | 61.1 | 62.2 | 60.1 | 63.0 | 57.9 | - | 34.2 | 40.2 | 41.2 | 38.5 | - | 34.6 | 46.1 | 43.7 | 41.5 |
| ScaIL | - | 38.1 | 43.6 | 51.7 | 51.9 | 53.2 | 53.1 | 55.6 | 58.4 | 59.3 | 56.5 | 52.1 | - | 36.1 | 36.8 | 40.2 | 37.7 | - | 39.8 | 42.6 | 50.9 | 44.4 |
| EEIL | - | 7.2 | 14.4 | 22.0 | 25.3 | 28.5 | 34.6 | 35.9 | 36.7 | 41.0 | 41.2 | 28.7 | - | 20.6 | 31.0 | 35.7 | 29.1 | - | 8.5 | 14.3 | 22.1 | 15.0 |
| FACT | - | 6.6 | 13.6 | 16.3 | 19.2 | 22.0 | 22.6 | 26.2 | 28.5 | 29.4 | 31.3 | 21.6 | - | 18.6 | 23.3 | 28.7 | 23.5 | - | 9.4 | 15.1 | 18.8 | 14.4 |
| Sem-aware | - | 13.8 | 17.5 | 22.0 | 21.6 | 21.8 | 24.1 | 27.7 | 27.9 | 29.0 | 31.5 | 23.7 | - | 16.5 | 23.5 | 27.5 | 22.5 | - | 13.2 | 19.8 | 22.6 | 18.5 |
| Microshape | - | 5.7 | 10.5 | 12.0 | 18.0 | 19.9 | 20.8 | 23.2 | 26.6 | 29.7 | 30.9 | 19.7 | - | 18.0 | 23.4 | 27.1 | 22.8 | - | 9.3 | 13.7 | 18.7 | 13.9 |
| C3PR | - | 4.3 | 6.9 | 9.8 | 12.9 | 13.5 | 15.9 | 18.8 | 22.4 | 23.3 | 24.4 | 15.2 | - | 14.3 | 20.0 | 23.2 | 19.2 | - | 7.9 | 10.7 | 14.9 | 11.2 |
| FoundationModel | - | **1.3** | 3.3 | 5.8 | 7.6 | **8.8** | 10.4 | 12.0 | **12.8** | **14.7** | 16.8 | 9.4 | - | 3.4 | 7.1 | 9.7 | 6.7 | - | 4.7 | 4.8 | 5.7 | 5.1 |
| **3D-FLEG(ours)** | - | **1.3** | **2.9** | **5.3** | **7.4** | 9.2 | **10.3** | **11.6** | 13.0 | 15.2 | **16.2** | **9.2** | - | **2.0** | **6.7** | **7.5** | **5.4** | - | **2.0** | **4.2** | **5.1** | **3.8** |

# A. Appendix for Experimental Results and Analysis.

To more accurately reflect the model's forgetting behavior during incremental learning, we further calculated the accuracy drop relative to the base classes at each incremental stage. The experimental results are presented in Tab. 5 and Tab. 6. These tables demonstrate that 3D-FLEG consistently exhibits competitive performance across every incremental stage. By embedding geometric information, our method effectively mitigates the forgetting of previously learned knowledge while efficiently learning new class representations with only a minimal number of new samples. This approach offers a novel solution for few-shot class-incremental learning on point clouds.

## B. Appendix for Hyperparameter Sensitivity

In our method, the update rate of the dynamic projection cluster in the incremental phase and the number of basis vectors contained in the dynamic projection cluster are two key parameters. In this session, we conducted experiments to investigate the impact of dynamically adjusting the update rate of projection clusters during the incremental learning phase on model performance. As illustrated in Fig. 6, setting an appropriate update rate during the incremental phase can further enhance model performance. In particular, when there are discrepancies between the data distributions of base classes and new classes, static base vectors, lacking an updating mechanism, gradually lose their representativeness and struggle to adapt to data changes, thereby affecting long-term performance stability. However, our experiments also revealed that choosing an appropriate update rate is critical. An excessively high update rate can lead to overfitting to the latest data, which may compromise overall model performance.

