# OpenReview forum: "Geometric Feature Embedding for Effective 3D Few-Shot Class Incremental Learning"
_ICML.cc/2025/Conference — ICML 2025 poster_

### Official Review · Reviewer_YEPj · 2025-03-03

**Overall Recommendation:** 3

**Summary:**

This paper investigates few-shot class incremental learning for 3D object classification using foundation models. Building on the work of FoundationModel (Ahmadi et al.), the authors employ a frozen, pre-trained large-scale 3D encoder (Uni3D) to extract generalizable features for each point. They then construct enhanced text embeddings based on prompts generated from category names, which are combined with geometric features to calculate similarity and assign the final label. A key contribution of the paper is the method for constructing abstract geometric features using spectral clustering and Laplacian eigenmaps, as well as the way to fuse the text embeddings from the prompts with the geometric features through transformer. Experimental evaluation across multiple datasets and settings demonstrates that the proposed method achieves clear improvements in performance.

**Claims And Evidence:**

The evidence provided to support the claims in the paper is insufficient in certain areas, as some details are lacking. For instance, the specific configuration used when removing a module in Table 3 is not clearly explained, leaving this aspect of the experiment unclear.

**Essential References Not Discussed:**

N.A.

**Experimental Designs Or Analyses:**

Yes, the overall experimental settings are valid, as they follow established protocols from existing methods. However, the ablation study in Table 3 lacks sufficient support, as the detailed configuration for removing a module is not clearly provided.

**Methods And Evaluation Criteria:**

Yes, the proposed method is evaluated both within-dataset incremental learning and cross-dataset incremental learning, in line with the protocols established by existing methods.

**Other Comments Or Suggestions:**

N.A.

**Other Strengths And Weaknesses:**

**Strengths**:

1. The proposed method is sound and well-constructed.
2. The paper demonstrates clear improvements across multiple datasets and FSCIL settings.
3. The ablation studies are comprehensive.


**Weaknesses**:

1. It would be beneficial to include visualizations of the basis vectors (and their evolution as new classes are introduced) or the distribution of geometric features.
2. Implementation details of the transformer encoder are not provided.
3. In Table 3, the detailed configuration for removing one module is not clearly explained.

**Questions For Authors:**

1. What principles guided the design of the five settings presented in Figure 1?

2. How does the observation in lines 87-93 lead to the conclusion that "one challenge to address to advance FSCIL on 3D point clouds" is "enhancing the model’s ability to learn robust feature representations"?

**Relation To Broader Scientific Literature:**

The key idea of this paper is closely related to the concept of cross-modal feature fusion, specifically integrating text and visual cues, which has been explored in prior research.

**Theoretical Claims:**

The paper does not present any theoretical claims.

---

> ### Author Rebuttal · Authors · 2025-03-31
>
> We sincerely appreciate your insightful feedback, which has guided us in refining the manuscript and addressing key concerns. Below, we provide detailed responses to each of your questions, supported by additional analyses and clarifications.
>
> **Q1: Clarification of Ablation Studies in Table 3**
> **A1**: Thank you for prompting us to clarify this section. Table 3 systematically evaluates the contributions of two critical components in the geometric feature extraction module:
> - **Dynamic Geometric Projection Clusters**: These clusters are constructed via spectral clustering and Laplacian eigenmaps to encode shared geometric structures.
> - **Attention Weights for Basis Vectors**: Learnable weights prioritize cluster centers relevant to incremental tasks.
>
> The ablation settings are:
> - Row 1: Raw point cloud features without DGPC.
> - Row 2: DGPC with equal weights for all basis vectors.
> - Row 3: DGPC + learnable attention weights.
> Key findings are now presented more succinctly:
> - DGPC Necessity: Removing DGPC (Row 1) degrades average accuracy by 2.7%, as raw features fail to integrate geometric-textual semantics.
>
> - Attention Mechanism: Adding learnable weights (Row 3) improves harmonic accuracy by 2.1% over Row 2, demonstrating adaptive weighting’s role in suppressing noise and outliers.
>
> The results validate that DGPC and attention weights synergistically enhance stability and discriminability. These results are now contextualized in Section 4.4, and additional visualization examples are available in Figure 2 of [the anonymous link](https://anonymous.4open.science/r/6061-FB07/6061.pdf). We previously did not express this clearly, but based on your feedback, we will restructure the table and discussion for better interpretability.
>
> **Q2: Visualization of Dynamic Projection Clusters**
> **A2**: To validate the effectiveness of dynamic geometric feature projection clusters during incremental learning, we visualize the geometric features extracted by DGPC at each incremental stage, as shown in Figure 2 of [the anonymous link](https://anonymous.4open.science/r/6061-FB07/6061.pdf). Visualizations reveal that DGPC-enhanced features exhibit tighter intra-class clustering and clearer inter-class separation. These results demonstrate that DGPC effectively encodes task-invariant geometric priors, enabling robust feature extraction across incremental phases. We have integrated key examples into the main text in our revised version.
>
> **Q3: Implementation Details of the Transformer Encoder**
> **A3**: The Transformer encoder comprises 2 standard layers, each with 8-head self-attention. We have enhanced the implementation details section in the revised version to include more comprehensive information.
>
> **Q4: Design Principles of Figure 1**
> **A4**: We sincerely appreciate your guidance in improving Figure 1's clarity. In light of your thoughtful suggestions, we have reorganized and optimized Figure 1 (detailed in Figure 1 of [the anonymous link](https://anonymous.4open.science/r/6061-FB07/6061.pdf)) to improve its clarity. It now contains 7 comparisons:
> 1. **SOTA Baselines**: Methods (1)-(2) adopt strategies from C3PR [1] and FoundationModel [2].
> 2. **Cross-Combination Strategies**: Methods (3)-(7) integrate various prompt styles with distinct training strategies.
>
> For detailed explanations of Figure 1, we sincerely invite you to consult our response to Reviewer 8rPH's Question 2.
>
> **Q5: Linking Observations to Robust Feature Learning**
> **A5**: We appreciate your guidance in strengthening this connection. The experiments (lines 87-93) demonstrate that complex prompt designs yield inferior performance compared to simple prompts. This observation highlights a critical limitation: existing methods overly rely on manually crafted text semantics while failing to autonomously extract geometry-aware robust features from 3D point clouds. Consequently, models exhibit excessive sensitivity to textual variations and struggle to adapt to distribution shifts in incremental phases with limited samples.
>
> 3D-FLEG addresses this by:
> 1. **Geometric Feature Embedding**: Explicitly encoding spatial structures into prompts via dynamic projection clusters, bypassing dependency on complex text engineering.
> 2. **Unified Optimization**: Forcing joint alignment between geometric features and text semantics during incremental training, enabling the model to prioritize discriminative cross-modal patterns from sparse data.
>
> By incorporating supplementary experiments, enhanced visualizations, and expanded explanations, we sincerely hope to have addressed all raised concerns. Please let us know if you feel any additional adjustments would better address your concerns.
>
> **References**
>
> [1] Canonical shape projection is all you need for 3d few-shot class incremental learning, ECCV 2024.
>
> [2] Foundation Model-Powered 3D Few-Shot Class Incremental Learning via Training-Free Adaptor, ACCV 2024.

---

### Official Review · Reviewer_g1xS · 2025-03-09

**Overall Recommendation:** 3

**Summary:**

The paper proposes 3D-GLEG, a method to improve 3D few-shot class incremental learning by incorporating geometric features into the learning process. The authors propose two modules: a geometric feature extraction module and a geometric feature embedding module. By leveraging geometric information, 3D-FLEG achieves superior performance on four datasets, ModelNet, ShapeNet, ScanObjectNN and CO3D.

**Claims And Evidence:**

1. The claim that Laplacian Eigenmaps can extract geometric structure from point cloud data is not well supported.

2. The claim that AdaptiveAvgPool1d for fine-grained feature extraction sounds not true for me. Based on my understanding, average pooling is not typically used for fine-grained feature extraction. Instead, it tends to extract global, smoothed features.

3. The claim that the geometric feature embedding module can ensure that data from both modalities interact on similar levels of abstraction is not fully supported according to section 3.4. How do equations 5 and 6 ensure that the point cloud features and text features have similar levels of abstraction? They are only in the same dimensional space. The authors only use a simple text prompt template that includes class names, which I believe contains abstract features. However, point cloud features should contain more detailed features.

**Essential References Not Discussed:**

N/A

**Experimental Designs Or Analyses:**

The paper evaluates the method on four datasets and uses multiple metrics. The results indicate improved performance over baseline methods. However, there are some potential issues:
1. No direct ablation study that analyzes the contribution of Laplacian eigenmaps or whether they truly enhance geometry-awareness and how.
2. More detailed discussion of why the method performs better across datasets.

**Methods And Evaluation Criteria:**

The proposed method and evaluation criteria are appropriate for the FSCIL problem. The evaluation on ModelNet, ShapeNet, ScanObjectNN and CO3D provides a solid benchmark, covering both real-world and synthetic 3D datasets. The metrics, including accuracy, harmonic mean accuracy, and relative accuracy drop, effectively measure both new class adaptation and forgetting mitigation.

**Other Comments Or Suggestions:**

1. It is confusing that both the ground-truth label and final feature representations computed from U are represented with the notation y.
2. Some of the Microshapes is spelled as "Micrpshapes"

**Other Strengths And Weaknesses:**

N/A

**Questions For Authors:**

1. For Figure 1, can the authors elaborate more on each prompt-training and training strategies and also add the citations if necessary? Since there are no detailed descriptions for the alignment module, how can the authors conclude that their method, embedding geometric features, is simpler than the alignment module?

2. Can the authors explain how they get the initial features, $featBase_i$, as they also mentioned the point cloud features $featPoint_i$?

**Relation To Broader Scientific Literature:**

For 3D FSCIL problem, previous works like Microshape proposed a universal description language to reduce domain discrepancies, and C3PR adapted CLIP to handle FSCIL task. This paper solves the problem by integrating geometric features, reducing reliance on foundation models and complicated training strategies.

**Theoretical Claims:**

The paper does not provide formal theoretical proofs but makes claims about the effectiveness of Laplacian eigenmaps in preserving geometric structures and the dynamic geometric feature projection clusters in improving feature representation. While the method is conceptually plausible, the paper does not rigorously prove that the transformed features are explicitly geometry-aware.

---

> ### Author Rebuttal · Authors · 2025-03-31
>
> Thank you for your insightful feedback, which has helped us significantly improve the clarity and rigor of our manuscript. Below are our detailed responses:
>
> **Q1: Claims (Laplacian Eigenmaps, AdaptiveAvgPool1d, geometric feature) are not well supported**
>
> **A1: (1) Laplacian Eigenmaps for Geometric Structure Extraction**
> Theoretically, Laplacian Eigenmaps minimizes $\sum_{i,j} (y_i - y_j)^2 W_{ij}$, where $y_i, y_j$ are low-dimensional representations of data points $x_i, x_j$, and $W_{ij}$ reflects their proximity ($W_{ij}=e^{-\frac{\|x_i-x_j\|^2}{t}}$ if neighbors; otherwise, $W_{ij}=0$). This ensures nearby points in the original space remain close in the reduced space.
> Using the graph Laplacian $L=D-W$, the problem reduces to minimizing $y^T L y$. The eigenvectors of $L$ capture the manifold's structure, preserving local geometry effectively. A similar theoretical analysis appears in [1].
>
> Besides, we have additionally conducted an ablation study to validate the role of Laplacian Eigenmaps as given below, demonstrating a 1.8% improvement in accuracy when utilized. We have incorporated the comprehensive theoretical proof and analysis in our revised paper.
>
> |Laplacian eigenmaps||Average|Accuracy|(%)|
> |:-:|-|-|-|-|
> ||Session0|Session1|Session2|Session3|
> |&cross;|**93.8**|91.2|86.4|85.0|
> |&check;|**93.8**|**91.9**|**87.5**|**86.8**|
>
> **A1: (2) AdaptiveAvgPool1d for Fine-Grained Features**
> We apologize for the confusion caused by the description of "fine-grained" features. You are right that traditional average pooling (AvgPool) typically extracts globally smoothed features. AdaptiveAvgPool1d in 3D-FLEG differs by dynamically adjusting pooling windows to capture local geometric statistics rather than fixed-window averaging. Specifically, it quantifies geometric attribute distributions across localized regions along the channel dimension, enabling fine-grained pattern extraction [2], where "fine-grained" refers to the statistical representation of local geometric details. We have revised our manuscript to clarify this distinction and articulate our design rationale.
>
> **A1: (3) Modality Abstraction Alignment**
> We have strengthened Section 4.3 to clarify the modality abstraction alignment:
> A dual-pooling strategy (Eq. 5) extracts multi-scale geometric features, bridging the gap between text and point cloud details. The Transformer encoder (Eq. 6) then refines these features, emphasizing geometry relevant to text prompts and reducing noise.
> Cross-entropy loss (Eq. 8) ensures consistency in the shared semantic space, aligning geometric and textual abstractions.
>
> **Q2: Cross-Dataset Superiority**
>
> **A2:** Thank you for prompting this critical analysis. 3D-FLEG’s cross-dataset superiority stems from its geometry-centric design:
> - **Dynamic Projection Clusters** capture task-invariant geometric patterns, which generalize across synthetic-to-real domains.
> - **Geometric Feature Embedding** directly fuses these features with text semantics, bypassing domain-specific text variations.
>
> This synergy enables 3D-FLEG to achieve 7% higher accuracy in cross-dataset tasks. We have revised Section 4.3 to clarify this mechanism.
>
> **Q3: Symbol Confusion and Misspellings**
>
> **A3:** Thank you for highlighting these inconsistencies. We have reviewed the manuscript and made corrections to the symbols and typographical errors to improve clarity and rigor.
>
> **Q4: Detailed Descriptions of Prompt and Training Strategies in Figure 1**
>
> **A4:** We appreciate your suggestion to improve Figure 1. We have detailed the experimental setup principles with relevant citations. For further details, please see our response to Reviewer 8rPH's Q2.
> The "Alignment Module and Dual Cache System" requires caching five samples per class, causing significant computational and memory overhead. In contrast, our geometric embedding module integrates geometric features directly with text prompts, eliminating the need for complex alignment training and caching.
>
> **Q5：Clarification on Initial Features ${featBase}_i$ and ${featPoint}_i$**
>
> **A5：** To address this ambiguity, we have revised Section 3 to explicitly define:
>
> ${featBase}_i$ : Features extracted from base-class data using the frozen Uni3D encoder. These features are used to construct dynamic geometric projection clusters.
> ${featPoint}_i$ : Features processed during incremental training from the same Uni3D encoder. These are dynamically reprojected through DGPC using learnable attention weights to extract geometric features.
>
> Your feedback has guided us in refining both the theoretical foundations and presentation of our work. All revisions have been incorporated into the manuscript.
>
> **References**
>
> [1] Laplacian eigenmaps for dimensionality reduction and data representation, Neural computation 2003.
>
> [2] Point cloud segmentation of overhead contact systems with deep learning in high-speed rails, Journal of Network and Computer Applications 2023.

---

### Official Review · Reviewer_8rPH · 2025-03-10

**Overall Recommendation:** 4

**Summary:**

The paper proposes a model called 3D-FLEG for the 3D few-shot class incremental learning task. The model has a geometric feature extraction module that obtains geometric features through clustering and Laplacian eigenmaps, and it includes a geometric feature embedding module to fuse these geometric features with text features, considering modality heterogeneity.

**Claims And Evidence:**

The claim that the reliance on text prompts and training strategies limits the robustness and performance of few-shot class incremental learning is reasonable and well supported.

**Essential References Not Discussed:**

The following papers are related to the paper's context and should be cited—including computing geometric features via clustering, improving generalization on novel classes, and fusing multimodal knowledge for novel class learning:

+ ICCV 2023, Generalized Few-Shot Point Cloud Segmentation Via Geometric Words
+ CVPR 2024, Rethinking Few-shot 3D Point Cloud Semantic Segmentation
+ ICLR 2025, Multimodality Helps Few-shot 3D Point Cloud Semantic Segmentation

**Experimental Designs Or Analyses:**

The main experiments are comprehensive and demonstrate the effectiveness of the proposed method. Nonetheless, some ablation studies are missing. For instance, the Geometric Feature Extraction Module uses spectral clustering as its first step, but the paper does not specify the number of clusters used in this step and analyze how varying this parameter affects performance. Additionally, it would be useful to know how the model's performance varies if the prompt style is changed, for example, to GPT-generated prompts.

**Methods And Evaluation Criteria:**

The proposed method is well explained, and the evaluation criteria and datasets used are appropriate for the task.

**Other Comments Or Suggestions:**

N/A

**Other Strengths And Weaknesses:**

The paper is clearly written and easy to follow. The motivations for the design choices are clear and reasonable.

**Questions For Authors:**

Please see the Experimental Designs part about the missing ablations. Providing the analysis on the mentioned ablation studies will further enhance the quality of the paper.

**Relation To Broader Scientific Literature:**

The proposed designs complement the broader literature and introduce new designs.

**Theoretical Claims:**

There are no theoretical claims.

---

> ### Author Rebuttal · Authors · 2025-03-31
>
> Thank you for your thoughtful and constructive feedback. We deeply appreciate your insights, which have guided us in refining our manuscript. Below, we outline the specific revisions made in response to your concerns:
>
> **Q1: Ablation Studies on the Number of Clusters**
> **A1**: We deeply appreciate your guidance on this critical aspect. Honestly speaking, we have added a detailed ablation analysis of the number of clusters in Table 6 of Appendix D (also given below), demonstrating how it affects model performance. Note that the cluster numbers are the same as the **Basis Vectors Count** in our experiments, as each cluster center is adaptively mapped to a corresponding basis vector within the projection space during dynamic projection cluster construction.
> As shown in Table 6, aligning cluster numbers with the base model's feature dimension (1024 in our case) balances information retention for accuracy and the redundancy of basis vectors. We have also included additional sensitivity analyses on cluster update rates as given in Fig. 6 of Appendix D.
> We have revised our manuscript to emphasize these findings more clearly in the main paper.
>
> |  |**Average**|      ||**Accuracy(%)**|**Harmonic**|**Accuracy**|**(%)**|
> |:---------------------:|------|------|------|------|-----|------|------|
> | **Basis Vectors Count**| 0    | 1    | 2    | 3   | 1    | 2    | 3    |
> | 256                 | 93.4 | 91.5 | 86.4 | 85.0 | 86.8 | 76.2 | 76.8 |
> | 512                 | 93.2 | 91.5 | 86.3 | 85.9 | 87.0 | 76.2 | 77.1 |
> | 1024                | 93.8 | 91.9 | 87.5 | 86.8 |87.0 | 77.4 | 77.5 |
> | 2048                | 93.6 | 91.6 | 87.5 | 86.7 |87.3 | 77.7 | 77.7 |
> | 4096                | 93.8 | 91.9 | 87.8 | 86.4 |87.0 | 77.7 | 78.3 |
>
>
>
>
>
> **Q2：Impact of Prompt Style on Model Performance**
> **A2**: Thank you for highlighting the importance of prompt-style analysis.
> We implemented a more comprehensive comparison of the 7 experimental configurations.
> They include:
> 1. **SOTA Baselines**: Methods (1)-(2) adopt strategies from C3PR [1] and FoundationModel [2].
> 2. **Cross-Combination Strategies**: Methods (3)-(7) integrate various prompt styles with distinct training strategies.
>
> From that, we recorded and provided a new Figure 1 (now visible in Figure 1 of [the anonymous link](https://anonymous.4open.science/r/6061-FB07/6061.pdf)) in our revised version.
>
> As shown in Figure 1, the performance of our geometric embedding (Method 7) remains stable across prompt variations, proving reduced dependency on prompt quality.
> This demonstrates that geometric feature embedding alleviates reliance on prompt quality by encoding structural priors, ensuring stable performance even under variations in prompt style.  Compared with "Alignment Module + Dual Cache System" (Method 2), which caches 5 samples per class, our strategy replays only a single sample and achieves 7% higher accuracy with lower memory overhead.
>
> **Q3：Cited related papers**
> **A3**: We sincerely appreciate the suggestion. The following works have been incorporated to strengthen our related works review:
> 1. **Geometric Word-Based Segmentation [3]**: Validates cluster-driven feature learning, aligning with our dynamic projection cluster design. (We have added it to Section 3.3 in our revised paper.)
> 2. **3D Few-Shot Generalization [4]**: Highlights domain adaptation challenges, motivating our geometry-centric approach for cross-dataset robustness. (We have added it to Section 3.4 in our revised paper.)
> 3. **Multimodal Fusion [5]**: Supports our cross-modal alignment strategy via joint geometric-textual optimization. (We have added it to Section 3.4 in our revised paper.)
>
> These references are now cited in relevant sections as indicated in the corresponding bracket.
>
> We hope our revised works have addressed your concerns.
> Should further clarifications or adjustments be needed, we are fully committed to incorporating your guidance.
> Thanks for your feedback, as it is invaluable in refining our work.
>
>
> **References**
>
> [1] Canonical shape projection is all you need for 3d few-shot class incremental learning, ECCV 2024.
>
> [2] Foundation Model-Powered 3D Few-Shot Class Incremental Learning via Training-free Adaptor, ACCV 2024.
>
> [3] Generalized Few-Shot Point Cloud Segmentation Via Geometric Words, ICCV 2023.
>
> [4] Rethinking Few-shot 3D Point Cloud Semantic Segmentation, CVPR 2024.
>
> [5] Multimodality Helps Few-shot 3D Point Cloud Semantic Segmentation, ICLR 2025.

---

> > ### Comment · Reviewer_8rPH · 2025-04-03
> >
> > Thank you for the detailed rebuttal. Now my concerns have been addressed and I would update my recommendation to accept.

---

> > > ### Author Response · Authors · 2025-04-04
> > >
> > > Dear reviewer,
> > >
> > > We sincerely appreciate your time and constructive feedback throughout the review process. We are delighted to hear that our rebuttal has addressed your concerns and that you now recommend acceptance.
> > > Your insightful comments have significantly strengthened our paper, and we are grateful for your valuable contribution to improving our work.
> > >
> > >
> > > Best wishes,
> > >
> > > All authors

---

### Decision · Program_Chairs · 2025-05-01

**Decision:**

Accept (poster)

**Comment:**

There are shared favorable opinions that the authors present solid and comprehensive evaluations and well-executed. The authors were enthusiastic about providing the requested ablation studies, and a couple of rounds of author-reviewer discussions led to a unanimous consensus that was positive. Note that multiple reviewers suggested that a theoretical analysis of the proposed method or module would be more beneficial in grasping the gist of the work.
Since no significant weaknesses were found, AC is glad to recommend acceptance of this hardworking paper.